# pH Distribution along Growing Fungal Hyphae at Microscale

**DOI:** 10.3390/jof8060599

**Published:** 2022-06-03

**Authors:** Bi-Jing Xiong, Claire E. Stanley, Christian Dusny, Dietmar Schlosser, Hauke Harms, Lukas Y. Wick

**Affiliations:** 1Helmholtz Centre for Environmental Research-UFZ, Department of Environmental Microbiology, Permoserstraβe 15, 04318 Leipzig, Germany; bijing.xiong@ufz.de (B.-J.X.); dietmar.schlosser@ufz.de (D.S.); hauke.harms@ufz.de (H.H.); 2Department of Bioengineering, Imperial College of London, South Kensington Campus, London SW7 2AZ, UK; claire.stanley@imperial.ac.uk; 3Helmholtz Centre for Environmental Research-UFZ, Department of Solar Materials, Permoserstraβe 15, 04318 Leipzig, Germany; christian.dusny@ufz.de

**Keywords:** bioreporter, microfluidics, hyphosphere, mycosphere, *Coprinopsis cinerea*, single cell

## Abstract

Creating unique microenvironments, hyphal surfaces and their surroundings allow for spatially distinct microbial interactions and functions at the microscale. Using a microfluidic system and pH-sensitive whole-cell bioreporters (*Synechocystis* sp. PCC6803) attached to hyphae, we spatially resolved the pH along surfaces of growing hyphae of the basidiomycete *Coprinopsis cinerea*. Time-lapse microscopy analysis of ratiometric fluorescence signals of >2400 individual bioreporters revealed an overall pH drop from 6.3 ± 0.4 (*n* = 2441) to 5.0 ± 0.3 (*n* = 2497) within 7 h after pH bioreporter loading to hyphal surfaces. The pH along hyphal surfaces varied significantly (*p* < 0.05), with pH at hyphal tips being on average ~0.8 pH units lower than at more mature hyphal parts near the entrance of the microfluidic observation chamber. Our data represent the first dynamic in vitro analysis of surface pH along growing hyphae at the micrometre scale. Such knowledge may improve our understanding of spatial, pH-dependent hyphal processes, such as the degradation of organic matter or mineral weathering.

## 1. Introduction

Fungi and bacteria co-inhabit a wide variety of environments. Creating unique and dynamic microenvironments, hyphal surfaces often allow for spatially distinct microbial interactions and functions near and/or affected by hyphae. The hyphosphere [1] is a zone of fungal activity and a favourable habitat [2,3] for bacterial colonisation and dispersal [4,5,6]. As well as oxygen availability [6,7], pH is an important driver for hyphal-bound microbial activity [1] such as bacterial motility [8]; degradation of organic matter, lignin and lignocellulose; the mobilisation and transport of nutrients [9,10]; mineral weathering [11]; and soil structure changes. Fungal mycelia typically modulate the environmental pH in their hyphosphere [12,13]. However, knowledge of the microscale pH of dynamically changing hyphal surfaces still is limited and often inferred only after destructive sampling [11]. pH analysis techniques involve nanoparticles [14,15], needle-type microelectrodes [16,17] or planar optodes [9]. Although nanoparticle-based sensors may allow pH profiles to be detected at the nanoscale, they are not yet commercially available. Despite being fast and non-invasive, planar optodes have a typical sensing resolution of >150 µm, and hence are not suitable for resolving pH at scales relevant for cellular functioning [18,19]. Needle-type microelectrodes with a tip size of ~3 µm [20] have been used to analyse microscale pH in growing hyphae. They are, however, highly invasive and, since used for mapping or time-lapse monitoring of pH along hyphae, would require 100 insertions per 1 mm to achieve a resolution of 10 µm. Many fungal mycelia, however, express significant multiscale heterogeneity in their extensive network and even between adjacent compartments of the same hypha [21], as has been evidenced by a spatially resolved fungal secretome analysis of *Aspergillus niger* mycelia [22]. Real-time microscale mapping of pH changes along extended mycelia using microelectrodes thus remains challenging. Using a whole-cell bacterial pH bioreporter (*Synechocystis* sp. PCC6803_peripHlu) that allows for spatial and temporal in vitro analysis of environmental pH at the single-cell scale (~3 µm), recent work [12] mapped spatially distinct and temporally stable gradients between pH 4.4 and 5.8 in habitats during hyphal colonisation. The pH bioreporter is based on a periplasm-localised, ratiometric pH-sensitive green fluorescent protein (GFP), pHluorin2, that displays varying bimodal excitation at 395 nm and 475 nm with maximum emission at 510 nm, thus reporting the local environmental pH [12]. By attaching >2400 *Synechocystis* sp. PCC6803 bioreporter cells to hyphal monolayers of the well-characterised saprophytic [23] basidiomycete *Coprinopsis cinerea*, we dynamically mapped the pH on the immediate surface of growing hyphae in a stringently controlled microfluidic system [24]. Time-lapse fluorescence microscopy was performed to examine pH-dependent fluorescence signals of individual bioreporter cells. While our study revealed an overall acidic mycelial surface, a temporally and spatially heterogeneous pH distribution along *C. cinerea* hyphae was demonstrated.

## 2. Materials and Methods

### 2.1. Microorganisms, Growth Conditions and Analysis of Water Contact Angles

*Synechocystis* sp. PCC6803_peripHlu was cultivated in modified blue-green 11 (BG 11) medium, as described before [12]. At the mid-exponential phase (OD_750_ = 2.8), 1 mL was harvested and centrifuged for 5 min at 7000× g at 10 °C, and the supernatant was discarded. The cell pellet was re-suspended in 5 mL of modified BG 11 medium to obtain a suspension of OD_750_ = ~0.05 that was used for further inoculation of the microfluidic devices (as described in Section 2.2.2). *C. cinerea* was used as a well-characterised pH-modifying filamentous fungus [23]. It was cultivated at 25 °C for 3 days on yeast–malt extract–glucose (YMG) medium [24].The water contact angles (θ_w_) of the fungus *C. cinerea*, and pH bioreporter *Synechocystis* sp. PCC6803_peripHlu were estimated using a drop size analyser (DSA) 100 system (Krüss GmbH, Hamburg, Germany). Briefly, mycelia of *C. cinerea* were cultivated for 2–3 days on filters with a diameter of 25 mm (0.45 μm, NC 45, Cellulose Nitrate Membrane Whatman, Maidstone, Kent, United Kingdom) placed on the surface of the YBG 11 [12] agar plate. *Synechocystis* sp. PCC6803_peripHlu cells were cultivated in flasks as described above and cell culture with an OD_750_ = 2.8 was placed to the aforementioned filters. Filters covered with either *C. cinerea* mycelia or the pH bioreporter cells were removed from the plate and placed onto filters with a diameter of 5 cm (pore size 0.45 µm) in a filtration unit, and washed three times under suction with 20 mL 10 mM phosphate-buffered saline (PBS), pH 7.2. Afterwards, filters were mounted on a glass slide and air-dried for two hours at room temperature. After that, contact angles were measured as detailed elsewhere [25,26].

### 2.2. Time-Resolved In Vitro Hyphal Surface pH Sensing

#### 2.2.1. Microfluidic Device

Microfluidic devices (Figure 1a) were prepared as described by Stanley et al. [24]. They were based on a channel architecture (Figure 1b) enabling laminar flow conditions [27] as a result of actively pumping solutions into the observation chamber. The key components of the device included (i) a fungal constriction channel, which limits the number of hyphae entering into the device and prevents backflow; (ii) an observation chamber allowing hyphal development and high-resolution microscopic imaging (i.e., Figure 1c); and (iii) a bacterial inlet (Figure 1b). The observation chamber (l × w: 5000 × 1000 µm) had a channel height of 10 µm to allow the formation of a monolayer of *C. cinerea* hyphae (hyphal diameter: ~7 µm). *Synechocystis* sp. PCC6803_peripHlu cells were introduced into the observation chamber by actively pumping cell solutions via the bacterial inlet (Figure 1b).

#### 2.2.2. Microfluidic Device: Fungal Inoculation and Loading of Bioreporters

Using a 1 mL syringe, the microfluidics were filled under sterile conditions with the modified BG 11 (pH 7.2) medium supplemented with 10 mM glucose for *C. cinerea*. Using a scalpel, a squared agarose piece (Ø: 0.5 cm) was cut from the peripheral growth zone of a *C. cinerea* plate. The piece was placed at a distance of ~1 mm from the entrance (Figure 1a,b) and the microcosm was sealed and incubated in the dark at 25 °C. The microcosms were examined regularly by microscopy to detect the time of hyphal arrival (after ca. 20 h of incubation) at the entrance of the observation chamber (denoted in the manuscript as *t* = 0 h). The setup then was incubated for another 18 h at 25 °C to enable the hyphae to cover two-thirds of the observation chamber (ca. 3500 µm, cf. Figure 1b). At *t* = 18 h, the microcosm was opened in a clean bench, and a sterile tubing system connected the inlet and outlet (Figure 1a,b) of the microfluidic. A suspension of pH bioreporter cells (OD_750_ = ~0.05, or 2.74 × 10^9^ cells L^−1^) was pumped into the observation chamber using a syringe pump (KD, Scientific Inc., North Logan, UT, USA) loaded with Luer-lock syringes (Injekt Solo, 2 mL, B. Braun, Melsungen, Germany) at a volumetric flow rate of 300 μL h^−1^ for 15 min in the clean bench. After that, the tubing system was disconnected and the microcosm sealed. The whole observation chamber was microscopically imaged to record mycelial structures and fluorescence signals of the locally distributed bioreporters on the hyphal surface. Mycelial growth and bioreporter signals were recorded for another seven hours at hourly intervals. The microcosms were kept in the dark when not used for microscopy.

Setups in the absence of a fungal inoculum were used as controls. In controls, a squared sterile agarose piece (Ø: 0.5 cm) was placed at a distance of ~1 mm from the PDMS stab. The subsequent treatment and microscopic monitoring of control microcosms were performed following the same procedure as described above, except the bioreporter loading rate was changed to 100 μL h^−1^ to increase bioreporter retention time in the observation chamber.

#### 2.2.3. Time-Resolved Microscopic Imaging

Ratiometric pHluorin2 expressed by *Synechocystis* sp. PCC6803_peripHlu displays a bimodal excitation at 395 nm and 475 nm with maximum emission at 510 nm [12,28,29]. Upon acidification, emission at 510 nm after excitation at 395 nm (I_510-395_) decreases, yet increases after excitation at 475 nm (I_510-475_). The 510 nm emission intensity ratio from two excitations (R_I510-475/I510-395_, abbreviated as R_I475/I395_) thus increases in response to decreasing environmental pH. Ratiometric fluorescence signals (I_510-475_ and I_510-395_) of individual bioreporter cells were monitored as described before applying minor modifications. Briefly, an automated inverted Zeiss microscope equipped with a Colibri LED fluorescence excitation unit (Axio Observer, Carl Zeiss Microscopy GmbH, Jena, Germany) and a customised fluorescence filter set was used for fluorescence imaging (AHF Analysentechnik AG, Tübingen, Germany). In time-lapse imaging, brightfield images were acquired using LED illumination (exposure time 80 ms, light source intensity 4.7 Volt). Fluorescence images were taken using an emission wavelength of 510 nm after 475 nm excitation (exposure time 100 ms, light source intensity 0.5 V) and 395 nm excitation (exposure time 200 ms, light source intensity 0.5 V). Both brightfield and fluorescence images were taken at hourly intervals. In fluorescence micrographs, red and blue were used as pseudo-colours to represent fluorescence emission excited at 475 nm and 395 nm, respectively. Our micrographs show the overlay of the two emission signals leading to pH-dependent changes of the bioreporter pseudo-colours, i.e., purple cells at pH 7, pink at pH 6 and red at pH 5, as described previously [12]. All microscopic images were taken at a total magnification of 400×, employing an objective lens Plan-Apochromat 40×/1.40 Ph3 M27 (Carl Zeiss Microscopy GmbH, Jena, Germany).

### 2.3. Image Analysis and Spatial Data Interpretation

The I_510-475_ and I_510-395_ of individual bioreporter cells loaded on the hyphal surface were analysed with ImageJ (https://imagej.net, accessed on 30 September 2021) [30], following a previously reported protocol [12]. The R_I475/I395_ of the individual bioreporter cells were calculated and transformed to pH values by using a previously established pH calibration curve (R_I475/I395_ = −0.5012 × (environmental pH) + 4.0386; validating range: pH 4.4 to pH 7.4 [12]). The time-resolved vector data (*n* > 2400 cells at each of the eight imaging time points) were then interpreted to examine (i) the average surface pH of the hyphae at given time points and (ii) the local surface pH of growing hyphae. The position of individual bioreporter cells is given by the longitudinal (*x*-axis) distance from the entrance of the microfluidic observation chamber (Figure 1b).

## 3. Statistical Analysis

Statistical analyses were performed using IBM SPSS Statistics software (version 26). The normality of data was verified with the Shapiro–Wilk test. To compare the differences between the two groups (i.e., pH at the hyphal tips and the more mature parts of the fungal mycelium), a *t*-test was used. To compare differences between multiple groups, means were compared using one-way ANOVA followed by either the least significant difference (LSD) test or Dunnett’s T3 test, depending on whether equal variances were or were not assumed, respectively.

## 4. Results

### 4.1. Time-Resolved Average Hyphal Surface pH

Using the pH-sensitive bioreporter *Synechocystis* sp. PCC6803 (Figure 2), we mapped the pH at the immediate surface of growing *C. cinerea* hyphae hourly (Figure 3) after the appearance of the first hyphal tips in the microfluidic observation chamber (ca. 18–25 h post inoculation, denoted as *t* = 0 h). At *t* = 18 h, the hyphal monolayer extended ~3600 µm into the liquid-filled observation chamber (Appendix A). Then, the pH bioreporter suspension was flushed through the observation chamber to attach the pH bioreporters to the hydrophobic (water contact angle *θ*_w_ = 128° ± 2°, Appendix A) hyphal surfaces (Figure 2). Only a few non-hyphal-bound cells (*n* < 50) were counted in the observation chambers. These cells were not included in the analysis of hyphal surface pH. We further observed some hyphal-attached pH bioreporter cells to disperse along *C. cinerea* hyphae via swarming or due to hyphal elongation (Appendix A, animated gif). This may explain why we also detected bioreporters on newly formed *C. cinerea* hyphae during the incubation (Figure 2b). Despite this re-distribution, bioreporter cell numbers remained quasi-stable throughout the experiment (2441 at *t* = 18 h and 2497 cells at *t* = 25 h).

Bioreporters revealed an average hyphal surface pH of 6.3 ± 0.4 at *t* = 18 h (Figure 3a; R_I475/I395_ = 0.88 ± 0.21, *n* > 2400 cells) and a subsequent drop to 5.0 ± 0.3 at *t* = 23 h (Figure 3a; R_I475/I395_ = 1.53 ± 0.14). Thereafter, the average pH remained unchanged up to the end of the experiment (*t* = 25 h, 5.0 ± 0.3, Figure 3a; R_I475/I395_ = 1.53 ± 0.15). The pH reported by the biosensors in fungus-free controls was 7.2 ± 0.1 and remained stable throughout the observation period (Figure 3a, triangles). Histograms reflecting the time-dependent pH frequency (Figure 3b) revealed substantial pH differences (∆pH) along mycelia ranging from ∆pH_18–20 h_ = 2.0–2.4 at *t* = 18–20 h and from ∆pH_22–25 h_ = ~1.6 at *t* = 22–25 h, as shown by the wider histogram range at *t* = 18–20 h compared to those at *t* = 22–25 h (Figure 3b).

### 4.2. Distribution of Surface pH along Growing Hyphae

Time-resolved vector data of pH sensed by the biosensors were analysed to examine pH distribution along the direction of hyphal growth (Figure 4). Different surface pH was observed along the growing hyphae at all observation points (Figure 4 and Appendix A). Surface pH at the hyphal tips (Figure 4) was found to be 0.7 (*t* = 18 h) to 0.9 (*t* = 25 h) pH units lower than at more mature hyphal areas near the entrance of the microfluidic observation chamber. Histograms of surface pH (Appendix A) at the hyphal tips (i.e., at a 500 µm distance away from the hyphal tips) showed higher pH differences (e.g., ∆pH = 2.9 at *t* = 18 h) than at the entrance (∆pH = 0.9 at *t* = 18 h) of the observation chamber (cf. Figure 4 and Appendix A).

## 5. Discussion

### 5.1. Time-Resolved In Vitro Analysis of Hyphal Surface pH

By combining a robust single-cell pH bioreporter and microfluidic technology, we developed a novel method allowing for dynamic high-resolution monitoring of spatial hyphal surface pH at micrometre scale. Our data reveal a dynamic spatial pH distribution in the direct vicinity of hyphal surfaces. In contrast to pH = 7.2 in a hypha-free environment (Figure 3a), we observed continuously decreasing average pH values around hyphae that can be ascribed to fungal metabolic activity. The pH approached a value of 5.0 ± 0.3 at *t* = 23 h. This finding is in line with an earlier study assessing the pH in the area surrounding hyphae of *C. cinerea* by abiotic and whole-cell bioreporter approaches [12], where an overall acid pH (pH 5.0) and temporally stable microscale pH gradients of ~1.4 pH units over distances of ~20 µm were detected in areas surrounded by *C. cinerea* hyphae. Using a microfluidic system to uniformly attach micrometre-sized (Ø: ≈3 µm) pH reporting *Synechocystis* sp. PCC6803 cells in a spatially specific manner, we report here microscale-sensed pH, i.e., at the immediate surface of growing hyphae.

Many studies report the attachment and biofilm formation of bacteria on fungal hyphae (e.g., [31,32,33]) as the basis of interactions such as mutualism, commensalism, antagonism or competition for nutrients or oxygen [31,34]. Force interactions and initial attachment of single bacteria [24,35] or other biological entities such as phages [32] to the fungal surface can be suitably described by the DLVO theory of colloidal interaction [36,37]. According to the DLVO approach, particle attachment to hyphal surfaces depends on both the van der Waals attraction (as can be approximated by the hyphal surface hydrophobicity and the water contact angle *θ*_w_) and the electrostatic repulsion between two surfaces in a liquid medium. Bacterial attachment to both hydrophilic and hydrophobic hyphal surfaces is thus to be expected. Here, we successfully loaded *n* > 2400 bioreporter cells to the *C. cinerea* growing hyphal monolayer (*θ*_w_ = 128° ± 2°) in a microfluidic device that allowed for non-invasive dynamic mapping of hyphal-attached bioreporter cells and their pH-sensitive signals (Figure 2 and Figure 3), as well as for the assessment of the assumed multiscale heterogeneity existing in filamentous fungi [21]. In our system, we observed poor attachment of bioreporter cells to the walls of the observation chamber in the presence of *C. cinerea* hyphae. As hyphal activity and bacterial fungal interactions may also depend on microscale gradients forming around hyphae, further development of our system will need to detect both surface pH and pH gradients forming around single hyphae, by developing tailored surfaces of the microfluidic observation chambers allowing for better bioreporter attachment.

### 5.2. pH on the Hyphal Surface and Its Ecological Significance

The litter decomposing fungus *C. cinerea* [23,38] is a well-characterised [23] lignocellulolytic basidiomycete. Basidiomycetes are known to excrete small organic acids, lowering the environmental pH in order to meet the requirements of their different extracellular enzymes [13,39,40] such as cellulases (optimal pH ~5 [41]), pectinases (optimal pH ~4 [42]) and phenol oxidases (optimal pH 4–5 [43]). In particular, the pH ranges observed in our study (6.3 ± 0.4 to 5.0 ± 0.3, Figure 3) seem to be in accordance with ideal pH levels reported for extracellular enzymes encoded in the *C. cinerea* genome [44,45]. For example, laccases involved in the degradation of lignin [44,45] and the oxidation of phenolic substrates are most active either at acidic (*C. cinerea* Lcc8 [46], optimal pH of 4.5–5.0) or circum-neutral pH (Lcc1 and Lcc9 [47,48], pH 6.5), respectively, whereas the optimal pH is 5–5.5 for lytic polysaccharide monooxygenases (LPMO) cooperating with laccases [49]. For dye decolourising peroxidases, optimal pH values of 3–6 were reported [2]. Although extracellular enzymes have been observed to diffuse a few hundred micrometres away from hyphae [50], many of them have also been found to be retained on the hyphal surface [51]. Thus, a favourable pH range on the hyphal surface may be one of the contributors to the high fungal activity observed close to the hyphal surface [11,52]. Likewise, such spatial and temporal pH traits may be used to dynamically identify the locations of high enzyme activity along hyphal surfaces. Using oxygen-sensitive particles (Ø: 8 µm), a recent study revealed microscale oxygen heterogeneity in the liquid (~10 µm [6]) around air-exposed hyphae of *C. cinerea* [6]. In our study, we observed 0.7–0.9 units lower and more variable pH values at growing tips than at the more mature hyphal parts (Figure 4 and Appendix A), thereby likely being a driver and result of hyphal elongation, as described by the acid growth hypothesis for plant root and filamentous organisms [20,53,54]. The acid growth hypothesis postulates that plant or fungal phytotoxin fusicoccin and plant hormone Indole-3-acetic acid can induce proton secretion concomitant with the induction of apical elongation by rapid acidification of the thick extension-limiting cell wall [55,56].

Although pH is known to be a major driver for microbial community structure [57,58], it remains mainly elusive as to whether, and to what degree, varying hyphal surface pH may influence the bacterial dispersal and/or the (uneven) colonisation of hyphae [59]. Hence, knowledge of dynamic pH distributions along mycelia could form an important next step in the examination of pH-mediated microniche differentiation on the hyphal surfaces in microbiome-on-a-chip [60] studies. As bacterial (flagellar) motility is known to be affected by the environmental conditions, e.g., the local pH [8,61,62,63], variations in hyphal surface pH may modulate bacterial swimming behaviour, and thereby influence bacterial transport along hyphal networks, and bacterial colonisation of and functioning in new habitats [5,64]. Spatially resolved microscale pH analysis with single-hypha resolution thus also enables a better understanding of bacterial fungal interactions and, hence, their roles in driving ecosystems services and functioning at the macroscale.

## Figures and Tables

**Figure 1 jof-08-00599-f001:**
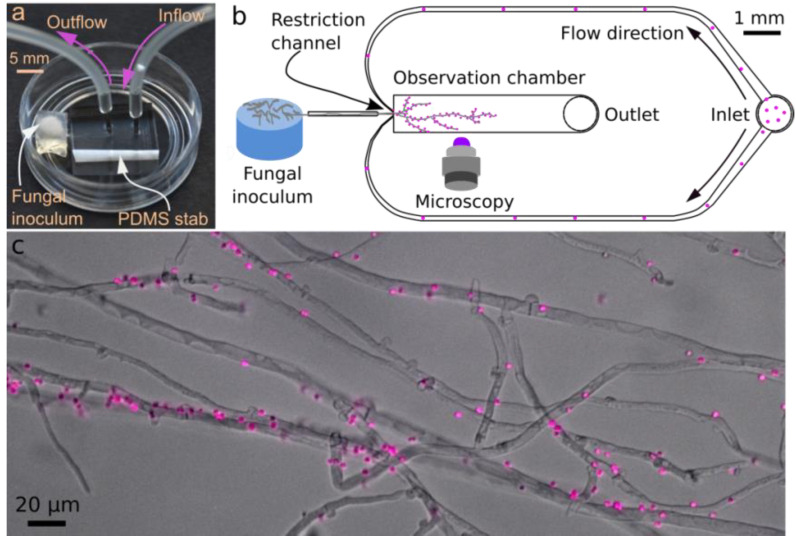
**Photograph and schematic of the microcosm for in vitro time-resolved pH monitoring at hyphal surfaces.** (**a**) Photograph of the experimental setup consisting of a fungal inoculum placed ca. 1 mm from the lateral opening of the microfluidic device and tubing used to load the bioreporter into to microchannels via the device inlet. (**b**) Schematic of the microcosm depicted in (a) consisting of an agar patch and microchannels embodied in a PDMS stab. The microchannels allow for the development of a hyphal monolayer in the observation chamber and subsequent loading of the pH bioreporters via an inlet and outlet system. (**c**) Micrograph showing typical distribution and attachment of *Synechocystis* sp. PCC6803_peripHlu pH bioreporters (pink dots) along hyphae of *C. cinerea* in the observation chamber.

**Figure 2 jof-08-00599-f002:**
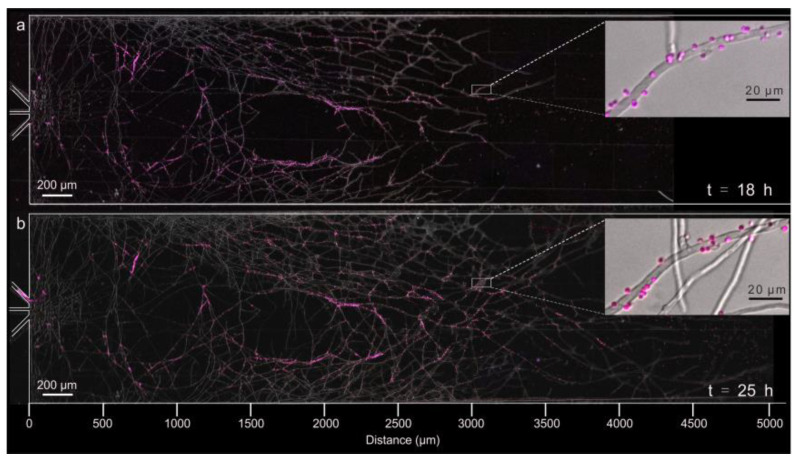
**Micrographs depicting hyphal development of *C. cinerea* in the observation chamber and corresponding changes in the pseudo-colours of hyphal-bound *Synechocystis* sp. PCC6803_peripHlu pH bioreporter cells.** The pseudo-colours refer to the overlay of two pH-dependent emission signals (R_I475/I395_). (**a**) Hyphal development at *t* = 18 h and most pH bioreporters show a pseudo-colour of magenta (pH ~ 6.3). (**b**) Hyphal development at *t* = 25 h and most pH bioreporters show a pseudo-colour of red (pH ~ 5.0). For better visibility, the contours of microchannels are marked by white lines.

**Figure 3 jof-08-00599-f003:**
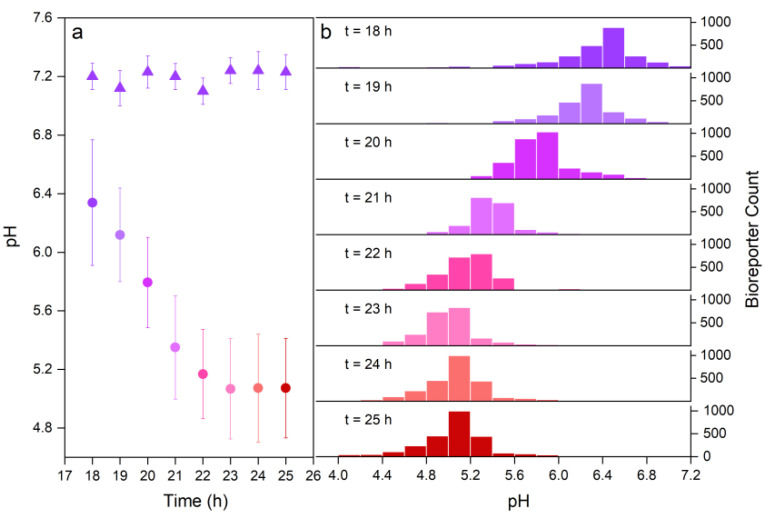
**Time-dependent average pH and histogram of pH distribution on hyphal surfaces of *C. cinerea*.** The pH was assessed by the hyphal-bound *Synechocystis* sp. PCC6803 bioreporter cells. (**a**) Bioreporter data encompass the average and standard deviation of *n* > 2400 cells (circles, (**a**)). Experiments in the absence of *C. cinerea* served as controls (triangles; *n* > 2200 cells). Time was denoted as *t* = 0 h when the first hyphal tips appeared in the microfluidic observation chamber. The bioreporter cells were loaded to the hyphae at *t* = 18 h of the noted time. (**b**) Corresponding pH distribution on hyphal surfaces of *C.*
*c**inerea* at *t* = 18–25 h.

**Figure 4 jof-08-00599-f004:**
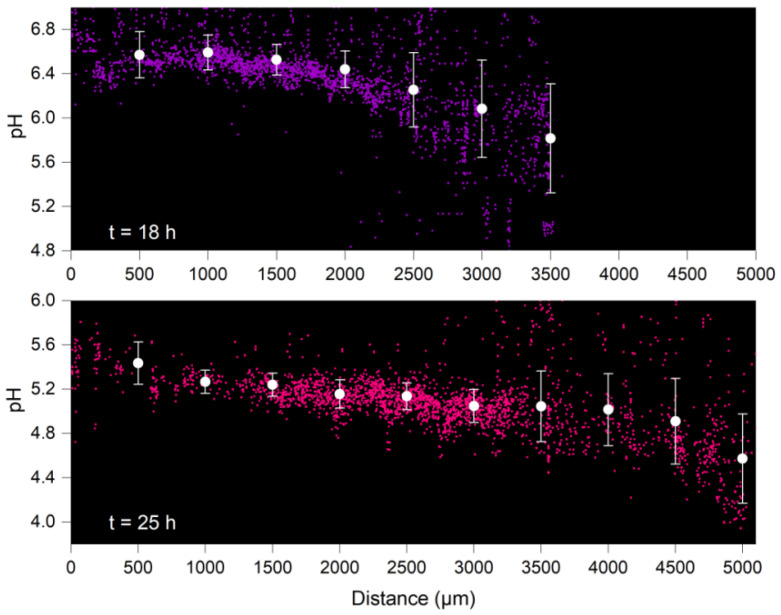
**Longitudinal distribution of sensed pH by *Synechocystis* sp. PCC6803 bioreporter cells attached to *C. cinerea* hyphae**. Data reflect average pH and pH distribution of *n* > 2400 cells incubated at various distances along the observation chamber at *t* = 18 h and *t* = 25 h. Average data include pH signals from all cells at ±500 µm from given distances. Significantly (*p* < 0.05) lower pH (≈0.8 pH) was observed near the hyphal tips (cf. at >3000 µm at *t* = 18 h, and at >4500 µm at *t* = 25 h) than those at the more mature part at the entrance of the observation chamber (x = 0–500 µm).

## Data Availability

The data presented in this study are available on request from the authors.

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
