# Peer review of "pH Distribution along Growing Fungal Hyphae at Microscale"

_jof, 2022, doi:10.3390/jof8060599_

Round 1

Reviewer 2 Report

The manuscript (ms) applies the pH-sensitive whole-cell bioreporters, developed by the group recently (Xiong et al., 2021, ISME Comm), for measuring pH distribution along growing fungal hyphae. The experimental-setup is well designed, and the results are supported with adequate figures. Also, the potential role of pH distribution in the activity of exoenzymes and in fungal-bacterial interactions are sufficiently discussed.

There is one bigger issue regarding the references, as they are numbered at the end of the ms from 1 to 65, but in the text of the ms numbers up to 86 are used. Please correct it, as in this way, the reader cannot check in the information in the referred articles.

Otherwise, the ms is suitable for publication after modifying the ms according to the following minor points:

· In Line 54 write “between pH 4.4 and 5.8” instead of “between pH 4.4 and 5.8 pH”

· On Fig. 2. and on Fig. S1 please also add the horizontal axe with distances from the entrance as on Fig. 4.

· On Fig. 3a I think that colours of the control samples (triangles) shouldn’t have to be changed so dramatically, as neither the pH of the control samples change.

· In Line 190 write “circles” instead of “cycles”.

Reviewer 3 Report

This manuscript reports the use of hyphae-attached biosensors and a microfluidic device to detect real-time microscale changes in pH. The development of this capability is very promising and will be significant for non-destructive characterization of ecological interactions and biogeochemical transformations at the micro-scale.  

However, the authors should provide further justification for their major claim that the biosensors are detecting dynamic changes in “spatial hyphal surface pH at micrometer scale.” (lines 214-215). Results in Figures 3, 4 and S2 support evidence of an overall change in pH across the entire microfluidic device. The biosensors may be attached to hyphal surfaces, but it seems possible that they could be responding to changes in pH in the surrounding media and not just pH changes at the hyphal surface. For example, in Figure 2, the insets show significant hyphal growth over 7 hours. Could these surrounding hyphae be acidifying the environment and influencing the biosensor’s fluorescent readout? Additionally, it is unclear to what extent organic acids could diffuse through the microfluidic device and influence biosensor read-out in other areas of the device. Judging from the pH variance above 3000 microns in Figure 4, it seems that there could be some micro-scale pH gradients occurring, but the reader would need to see this represented as a microscopy image to determine whether this variance is along a single hyphae or across the device’s hyphal network.

Additional Comments:

-Please fix the bibliography as some references appear to be missing or incorrect. The manuscript cites up to ref. 86 but the bibliography only goes to ref. 65. Ref. 13 and Ref 25 appear to be the same journal article. Ref. 31 is cited as a source for pH reporter pseudo-colors but it references ImageJ, many references only have one author… etc.

-Please explain the mechanism for how Synechocystis cells were able to attach to newly grown hyphae over the 7 hours of observation. For example, in Figures 4 and S2, no biosensor was detected past 3500 microns at t =18 but, at t=25h, biosensor was detected up to 5000 microns along the microfluidic chamber. Do biosensor cells replicate and move to a new location along the hyphae or does the hyphae elongate with the attached biosensor cells?
